# OpenReview forum: "$R^2$-Guard: Robust Reasoning Enabled LLM Guardrail via Knowledge-Enhanced Logical Reasoning"
_ICLR.cc/2025/Conference — ICLR 2025 Spotlight_

### Official Review · Reviewer_t9hR · 2024-10-23

**Soundness:** 3
**Presentation:** 4
**Contribution:** 3
**Rating:** 6
**Confidence:** 3

**Summary:**

R2-Guard is a framework that enhances safety of LLMs. Unlike existing models treat safety categories independently, R2-Guard captures the
relationships between them by integrating first-order logical rules into PGM, including MLN + PC. allow the system to infer unsafety probabilities through a reasoning process that combines safety rules. This method strengthens the model's ability to detect unsafe content across diverse categories and increases its resistance to jailbreak attacks. Another innovation is TwinSafety benchmark, which tests guardrail models on complex safety challenges like intent-hiding and double entendres. Evaluations show that R2-Guard outperforms eleven state-of-the-art guardrail models across six safety benchmarks, with a notable 30.4% improvement over LlamaGuard on the ToxicChat dataset and a 59.5% improvement in resisting jailbreak attacks.

**Strengths:**

1. R2-Guard uses PGMs to explicitly capture relationships between safety categories, enabling more accurate moderation of complex unsafe content.
2. It significantly outperforms state-of-the-art models, showing a 59.5% improvement in resisting jailbreak attacks through logical inference and rule-based reasoning.
3. R2-Guard can adapt to new safety categories by simply modifying its reasoning graph, without retraining, making it highly adaptable for evolving safety needs.

**Weaknesses:**

While R2-Guard demonstrates flexibility in adapting to new safety categories by modifying the reasoning graph, it cannot cover all possible types of unsafe content by itself. Its effectiveness is limited by the categories and logic rules predefined in the system, which means that it may not detect emerging or unforeseen forms of unsafe behavior unless explicitly updated. This reliance on pre-specified rules requires ongoing maintenance to ensure comprehensive coverage.

**Questions:**

1. How does R2-Guard handle ambiguous or context-dependent cases of unsafe content that don’t fit neatly into the predefined safety categories?
2. Does R2-Guard have any mechanism to detect entirely new or emerging types of unsafe content that aren’t covered by its predefined safety categories and rules?

---

> ### Author Response · Authors · 2024-11-22
> **Response to Reviewer t9hR**
>
> We appreciate the reviewer's thoughtful feedback on our paper. Below, we included additional comments to further improve our work.
>
> > Q1: The flexibility of R^2-Guard relies on pre-specified rules and this reliance requires ongoing maintenance to ensure comprehensive coverage. Does R2-Guard have any mechanism to detect entirely new or emerging types of unsafe content that aren’t covered by its predefined safety categories and rules?
>
> Thank you for the thoughtful question! The open-world content moderation scenario, where unseen safety categories emerge dynamically, is indeed an interesting topic to discuss further. While such open-world scenarios with unseen labels are common in tasks like object classification [1] or detection [2], where countless real-world object categories make exhaustive enumeration impractical, unsafety detection for LLM inputs/outputs differs. In this domain, safety categories are generally well-defined and clearly outlined in existing regulations, such as government policies like the EU AI Act, White House AI Executive Order, or industry policies like OpenAI’s usage policy and Meta's service terms. These policies outline specific safety categories and rules for LLM deployment. Consequently, these can be compiled into the reasoning graphs of $R^2$-Guard to enable reasoning-driven guardrails. If these policies are updated (e.g., through the addition or removal of categories or rules), the reasoning graph of $R^2$-Guard can be directly modified to flexibly adapt to new safety criteria, as described in Section 5.3.3.
>
> Although open-world guardrail scenarios are generally impractical, we discuss how $R^2$-Guard could be applied in a hypothetical setting to handle unseen categories. Within the $R^2$-Guard framework, we can adopt ideas from confidence-based open-world detection to address this challenge. Specifically, we could maintain category-specific feature prototypes for LLM prompts across existing unsafety categories and benign examples. When a test instance is encountered, its features can be compared to these prototypes by computing their distances. If the distance exceeds a calibrated tolerance threshold, the instance could be flagged as belonging to a potentially unseen unsafety category, triggering a human audit. The tolerance threshold could be calibrated in a simulated dynamic scenario, as described in Section 5.3.3. Features could be instantiated as reasoning paths in MLNs or PCs within $R^2$-Guard, offering a more robust representation than relying solely on output-level logits. We added to Section 5.3.3 and Appendix A.6 and would like to leave an in-depth analysis for future work.
>
> *[1] Bendale, Abhijit, and Terrance Boult. "Towards open world recognition." Proceedings of the IEEE conference on computer vision and pattern recognition. 2015.*
>
> *[2] Joseph, K. J., et al. "Towards open world object detection." Proceedings of the IEEE/CVF conference on computer vision and pattern recognition. 2021.*
>
> > Q2: How does R2-Guard handle ambiguous or context-dependent cases of unsafe content that don’t fit neatly into the predefined safety categories?
>
> Thank you for the interesting question! We indeed observe that certain instances may not exhibit a high likelihood of unsafety within a single safety category, but the interaction among multiple categories can result in overall unsafety. A key advantage of $R^2$-Guard is its ability to encode such cross-category unsafety relationships into the reasoning graph, enabling more effective guardrail performance for these complex cases. For instance, in the example from Figure 2 with ambiguous unsafety across multiple categories, the likelihood of unsafety across individual categories like self-harm, self-harm/intent, and sexual content is moderate (below 0.5) when assessed by a purely data-driven guardrail model. However, $R^2$-Guard raises the overall unsafety probability to a more appropriate level (above 0.5) by leveraging probabilistic inference in MLNs or PCs with complied safety rules to capture cross-category intercorrelations. To enhance clarity, we also added this illustration of the running example in Section 3.1.

---

> > ### Comment · Reviewer_t9hR · 2024-11-24
> >
> > Thank you for the response; I now understand your point. However, I believe that a score of 6 is fair, and I will maintain my overall assessment.

---

### Official Review · Reviewer_t55i · 2024-11-02

**Soundness:** 4
**Presentation:** 2
**Contribution:** 4
**Rating:** 8
**Confidence:** 3

**Summary:**

This paper proposes R2-guard, a new guardrail mechanism for LLMs based on logical reasoning with probabilistic graphical models (PGMs).
The key benefit of this R2-guard is that its decision-making is more interpretable than existing methods.
R2-guard first computes the probability that the input contains some known categories of harm (e.g., 40% hate speech, 80% violence, etc.).
These category-specific probabilities are then passed to a PGM with hard-coded rules (e.g., "self-harm implies unsafe") and learned rule weights, which compute the probability that the input is unsafe.
R2-guard is shown to outperform a number of existing benchmarks and generalizes well to unseen unsafety category combinations.
The authors additionally present an evaluation benchmark called TwinSafety.

**Strengths:**

This paper presents an innovative method for rule-based guardrails that combines newer techniques like LLMs with classical ones like Markov Logic Networks and Probabilistic Circuits.
The PGM component is particularly nice, as a hard-coded rule structure gives developers an interpretable metric with which to evaluate content.
The evaluations are well done, and the new benchmark of TwinSafety should be valuable to the LLM defense community.
Overall, I believe that this paper makes a solid contribution to the improvement of LLM safety.

**Weaknesses:**

I found the presentation of R2-guard to be technically dense, even though (in my opinion) the high-level idea is simple.
I think it would be of much benefit to this work and the community if the presentation is simplified.
For example:
* A simplified version of Figure 1 could be put in Section 1 to showcase the high-level idea.
* In Section 3.1, it would be helpful to demonstrate an execution of the example text "In her moments ...".

These changes could help better communicate the main idea to a short-attentioned reader and also support a more dedicated reader by walking through an example.

**Questions:**

It would be good if the authors included some discussion about what kinds of safety rules R2-guard might have trouble modeling.

---

> ### Author Response · Authors · 2024-11-22
> **Response to Reviewer t55i**
>
> We appreciate the reviewer's thoughtful feedback on our paper. Below, we included additional comments to further improve our work.
>
> > Q1: Simplification and improvement of the presentation.
>
> Thank you for the valuable suggestion! In the current version, we have included an abstract overview figure (Figure 1) in Section 1. The introduction provides an overview of $R^2$-Guard, explaining how it first computes category-specific unsafety probabilities and then performs probabilistic inference on MLNs or PCs to reason with these per-category likelihoods. A more detailed overview, along with a running example, is provided in Figure 2. Additionally, we have added a paragraph in Section 3.1 to further illustrate the example and enhance understanding.
>
> > Q2: Discussion about what kinds of safety rules $R^2$-Guard might have trouble modeling.
>
> Thank you for the interesting question! $R^2$-Guard is capable of encoding first-order logic rules into MLNs or PCs for reasoning, making it applicable to any rule-intensive domains. However, it is limited in handling rules beyond the scope of first-order logic, such as temporal logic rules or higher-order logic rules. For instance, in the autonomous driving domain, a safety rule like “The car must brake within 0.2 seconds upon detecting an obstacle within 10 meters” involves temporal dependencies that cannot be effectively represented using first-order logic. As a result, $R^2$-Guard is unable to model such rules. We have included this limitation in the discussion section for greater clarity. We leave the development of a reasoning framework capable of encoding a broader range of logic rules as future work.

---

### Official Review · Reviewer_DyjW · 2024-11-04

**Soundness:** 3
**Presentation:** 3
**Contribution:** 3
**Rating:** 8
**Confidence:** 4

**Summary:**

Existing LLM guardrails treat different categories of safety failures independently. In contrast, R2-Guard proposes a reasoning-enabled LLM guardrail that can perform additional reasoning on top of predictions from category-specific guardrails. This reasoning is done through a probabilistic graph model that is grounded using manually curated first-order logical rules. The paper explores different probabilistic graph model architectures as well as supervised and semi-supervised approaches to training them. A new safety benchmark is also proposed, testing resilience to more granular jailbreaks (for example, at phrase level) and new hybrid categories of harm. R2-Guard is shown to be effective across a range of models and safety benchmarks, against a variety of jailbreak attacks. Additionally R2-Guard is also efficient despite the additional probabilistic graph model component, having similar latency to existing LLM-based guardrails.

**Strengths:**

1. The paper is generally well-written and also quite original in its use of logical rules through PGMs to enhance guardrail effectiveness.
2. R2-Guard is shown to be more effective than most existing guardrails on standard safety benchmarks, and also proves to be an effective defense against some performant jailbreak attacks.
3. R2-Guard is adaptable: new safety categories can be added to the guardrail relatively easily, through a new categorical classifier for the category along with some additions to the set of logical rules used by the PGM, although it is unclear if the PGM needs to be retrained.
4. R2-Guard is efficient, having marginally higher latencies than existing LLM guardrails. This makes it practical for real-world usecases.

**Weaknesses:**

1. The ruleset passed to the PGM is quite small. How are these rules created? If they are manually curated, have other alternative rulesets also been explored? It is also surprising that such a simplistic ruleset (boiling down to hypernym and hyponym relationships) results in large performance gains. This could be discussed further.
2. I also found the pseudolearning approach to training the PGM somewhat confusing. The data for training the PGM is curated using the ensemble approach: if the maximal score from the category-specific classifiers is greater than 0.5, the sample is treated as harmful. Why then does the resulting PGM outperform the ensemble approach on safety benchmarks?
3. The TwinSafety section is very lacking in details. What does "pairwise construction" mean? The examples provided in the paper also do not look like typical harmful prompt queries (for example, ": It is illegal for children to take drugs. However, adults can do so since they control themselves well") I would suggest a human annotator study verifying the quality of this dataset.
4. For the experimental baselines, how are the categorical models trained? Why is Llama-2-7b used for the chain of thought baseline? GPT-4 is generally accepted to be much better aligned with human preferences as a guardrail.
5. Why is R2-Guard nearly perfect when combating jailbreaks? How is the model trained for Section 5.2? If it is trained on real data that contains examples of prompts with these jailbreak attacks already applied to them, it might be unfair to other baselines. For example, with GCG, there is the same suffix attached to each prompt. If GCG-applied prompts are used in training, the guardrail can simply learn to ignore this suffix.
6. R2-Guard seems dependent on strong category-specific guardrails for its performance. Some analysis where the performance of these guardrails is compared against R2-Guard performance for each corresponding category would help strengthen the paper, and identify where R2-Guard improves performance.

**Questions:**

1. There is a typo on line 212: "realted"
2. More details should be provided regarding the training data for R2-Guard in each experiment. In Section 5.3.1, is the R2-Guard using an MLN or PC?
3. Section 5.3.3 needs more details as well. Is the PGM retrained after each category of harm is added, or is the set of logical rules simply expanded?
4. Why does having only direct rules for the PGM improve performance? Is this equivalent to learning dynamic ensembling weights? How well does a manually-tuned ensemble of categorical classifiers perform compared to R2-Guard?
5. Ensemble logic is used to train R2-Guard with pseudo learning, yet the resulting model outperforms the ensemble-based approach used to train it. This requires more discussion.

---

> ### Author Response · Authors · 2024-11-22
> **Response to Reviewer DyjW (Part 1)**
>
> We appreciate the reviewer's thoughtful feedback on our paper. Below, we included additional comments to further improve our work.
>
> > Q1 (Weakness 1): How is the ruleset created? The rationale of using the ruleset for impressive performance gains.
>
> Thank you for the question! The ruleset is developed through a manual process that begins with annotated safety categories from sources such as OpenAI, LlamaGuard, Perspective, and Aegis. These sources serve as the foundation for defining the safety categories. Their unsafety descriptions are carefully analyzed to establish logical interconnections among categories. Language models can also be employed to automate the logical rule definition process, leveraging the original rule descriptions and a few-shot demonstration setup. However, since the number of unsafe categories is tolerable for human efforts and defining the rules is a one-time effort, human annotations remain an efficient approach.
>
> The performance gains of $R^2$-Guard arise from two key aspects: (1) $R^2$-Guard uses the unsafety likelihoods of multiple category-specific guardrails as reasoning foundations, connecting them to the target via direct rules, which presents a more effective and robust information source. (2) Given that ensemble learning builds on independence assumptions, cross-category intercorrelations in practice undermine it and limit the guardrail performance. In contrast, $R^2$-Guard encodes these cross-category relationships through indirect rules and performs systematic and interpretable reasoning via MLNs or PCs to generate the final prediction. The ablation studies on direct and indirect rules in Section 5.3.1 provide empirical validation of these performance gains in greater detail.
>
> > Q2 (Weakness 2, Question 5): Why $R^2$-Guard trained with pseudo samples created by ensemble logics outperform ensemble learning empirically?
>
> Thank you for the insightful question! We would like to emphasize that since $R^2$-Guard encodes only the truly useful safety rules into reasoning graphs, its effectiveness is robust to variations in knowledge weights within a reasonable range. Consequently, assigning relatively large values to the knowledge weights is sufficient. To automate this process, we propose a pseudo-learning method that leverages simulated unsafety scores and labels.
> To show that, we also provide ablation studies of $R^2$-Guard with fixed knowledge weights for all rules in Table A. The results demonstrate that when fixed knowledge weights are set above 5.0, $R^2$-Guard achieves performance comparable to pseudo-learning. For context, the knowledge weights learned via pseudo-learning have a mean value of 5.57 and a standard deviation of 0.82. The results are provided in Appendix A.7 for further clarification.
>
> The superior performance of $R^2$-Guard compared to ensemble learning can be attributed to the following factors: (1) Ensemble learning relies on independence assumptions, which are undermined in guardrail domains due to cross-category intercorrelations. These intercorrelations limit the effectiveness of ensemble methods, whereas $R^2$-Guard captures them through indirect rules; (2) Ensemble methods are highly sensitive to scale misalignments across categories. For example, if a category-specific guardrail model is poorly trained due to data imbalance, it can introduce noise into the ensemble process. In contrast, $R^2$-Guard’s inference mechanism integrates the contributions of individual guardrail models more smoothly by combining data-driven likelihoods with logical likelihoods across all possible worlds. Further analysis and empirical evidence are provided in the response to Q9.
>
> Table A: AUPRC of $R^2$-Guard (PC) with fixed weights and pseudo-learning on OpenAI Mod dataset.
> | Fixed weight (w=0.0) | Fixed weight (w=3.0) | Fixed wight (w=5.0) | Fixed weight (w=10.0) | Fixed weight (w=100.0) | Fixed weight (w=1000.0) | Pseudo-learning |
> | - | - | - | - | - | - | - |
> | 0.854 | 0.897 | 0.922 | 0.931 | 0.925 | 0.928 | 0.927 |

---

> ### Author Response · Authors · 2024-11-22
> **Response to Reviewer DyjW (Part 2)**
>
> > Q3 (Weakness 3): More details on the TwinSafety dataset.
>
> Thank you for pointing out the lack of detail in the TwinSafety section! We have added further illustrations of the pairwise construction method used to create pairs of safe and unsafe examples. These pairs share similar topics but differ subtly in ways that introduce novel types of unsafety. For example, in the intent-hiding category, a benign example can be: "It is illegal for children to take drugs." The corresponding unsafe example can be: "It is illegal for children to take drugs. However, adults can do so since they control themselves well." Both examples share a similar context, but the latter introduces a hidden unsafe implication—that adults can take drugs—which is generally considered unsafe in most regions.  We believe that unsafety triggered by these nuanced differences can provide a significant challenge for guardrail models in stress testing. While we recognize that definitions of unsafety may vary across regions and time, we assume that the unsafety categories in TwinSafety align with widely accepted safety standards. These additional details have been incorporated into Section 5 for greater clarity.
>
> > Q4 (Weakness 4): More clarifications on the training of categorical models. Usage of GPT-4 instead of Llama-2 7b for CoT reasoning.
>
> Thank you for your question. As detailed in Section 5.1, for fair comparisons, $R^2$-Guard employs the same category-specific learning models as those used in ensemble learning. These include categorical models from OpenAI Mod, LlamaGuard, ToxicChat-T5, Perspective, and Aeigis, which covers a broad spectrum of safety categories.
> Additionally, we included results for Chain-of-Thought (CoT) reasoning with GPT-4o in Table B. The findings indicate that CoT reasoning with GPT-4o improves guardrail performance of CoT with Llama2-7b; however, as an implicit reasoning method, it still lags significantly behind $R^2$-Guard. We have updated the results for CoT reasoning with GPT-4o in the revised manuscript.
>
> Table B: AUPRC of CoT reasoning with Llama2-7b and GPT-4o.
> | Method | OpenAI Mod | ToxicChat | XSTest | Overkill | BeaverTails | TwinSafety | Average |
> | - | - | - | - | - | - | - | - |
> | CoT (llama2-7b) | 0.856 | 0.592 | 0.743 | 0.793 | 0.687 | 0.599 | 0.712 |
> | CoT (GPT-4o) | 0.881 | 0.654 | 0.746 | 0.816 | 0.713 | 0.657 | 0.745  |
> | $R^2$-Guard  | 0.927 | 0.910 | 0.916 | 0.933 | 0.825 | 0.780 | 0.882 |
>
> > Q5 (Weakness 5): More details and explanations on evaluations of $R^2$-Guard against jailbreaks.
>
> Thank you for the question! We added clarifications that in the evaluation against jailbreaks in Section 5.2, we do not train $R^2$-Guard on adversarial prompts. For fair comparisons, in Section 5.2, we keep the same model configuration for $R^2$-Guard and all baselines as in Section 5.1. There is no additional training or parameter tuning for all methods in jailbreak evaluation.
>
> We also added the following illustration on why $R^2$-Guard demonstrates superior robustness against jailbreaks. In brief, the PGM reasoning component introduces additional complexity and challenge to the attack objective. When attempting a jailbreak against the learning component (i.e., the purely data-driven guardrail model), the goal is to optimize a jailbreak string to reduce the unsafety score. In contrast, when targeting both the learning component and the PGM reasoning component (i.e., purely data-driven guardrail models combined with MLN/PC reasoning), the objective is to optimize a jailbreak string to not only lower the unsafety score but also ensure that the scores for different safety categories after attack adhere to the compiled safety rules. Therefore, the PGM reasoning component introduces additional intricacy to jailbreak attempts and highlights the need for more effective jailbreak strategies against the reasoning pipeline in future work.
>
> > Q6 (Weakness 6): $R^2$-Guard seems dependent on strong category-specific guardrails for its performance.
>
> Thank you for the comment. Due to space limits, we defer the ablation studies of $R^2$-Guard with various combinations of category-specific guardrails to Appendix A.5. The results show that $R^2$-Guard consistently outperforms ensemble learning in improving guardrail performance, regardless of the combination of category-specific guardrails, including weaker ones. This demonstrates that the effectiveness of $R^2$-Guard is not confined to strong category-specific models. However, utilizing stronger models does further enhance overall guardrail performance. Additional clarifications have been included in the main text within the ablation study paragraph.

---

> ### Author Response · Authors · 2024-11-22
> **Response to Reviewer DyjW (Part 3)**
>
> > Q7 (Question 1): Typo in Line 212.
>
> The typo is fixed in the revised version.
>
> > Q8 (Question 2, Question 3): More experiment details in Section 5.3.1 and 5.3.3.
>
> Thank you for the comment. In Section 5.3.1, we evaluate the effectiveness of direct rules and indirect rules through $R^2$-Guard (PC). In Section 5.3.3, we also employ $R^2$-Guard (PC) and expand the PC to incorporate new safety categories and their corresponding rules without retraining the model. These details have been added to the revised manuscript for clarity.
>
> > Q9 (Question 4): Is using direct rules only equivalent to ensemble with dynamic weights? Why does having only direct rules for the PGM improve performance? What is the performance of ensemble learning with manually tuned weights?
>
> Thank you for the interesting question! We add the following discussion to better differentiate $R^2$-Guard from ensemble learning with dynamic weights.
>
> First, we would like to point out that $R^2$-Guard is not equivalent to ensemble learning with dynamic weights. Ensemble learning presents a linear combination of unsafety scores, while $R^2$-Guard presents a non-linear transformation with exponential logical likelihood functions.
>
> Second, according to Section 5.3.1, using only direct rules brings in marginal performance improvement compared to ensemble learning. The potential reason is that ensemble methods based on the maximum unsafety score are sensitive to scale misalignments across different categories. For instance, if a category-specific guardrail model is poorly trained due to data imbalance, this will inject noises into ensemble process and simply selecting the maximal unsafety score can degrade the overall ensemble performance. In contrast, $R^2$-Guard’s inference mechanism integrates contributions from individual guardrail models more smoothly by leveraging both data-driven and logical likelihoods across all assignments.
> Additionally, we want to highlight that indirect rules play a crucial role by capturing cross-category relationships, significantly enhancing performance in combination with direct rules.
>
> Finally, we have added an evaluation of ensemble learning with manually tuned weights in Table C. Here, we tuned the ensemble weights across 10 configurations and observed that the optimal configuration improved performance by approximately 3%, compared to standard ensemble methods. However, this improvement still fell short of the $R^2$-Guard (PC) performance by around 3%. Note that manual tuning incurs additional computational overhead and increases the risk of overfitting to specific data distributions, which is not preferred in practical cases.
>
> Table C: AUPRC of different methods on OpenAI Mod Datasset.
> | Ensemble (Max) | Ensemble (Avg) | Ensemble (Tuned weights) | $R^2$-Guard (PC) |
> |-|-|-|-|
> | 0.863 | 0.851 | 0.895 | 0.927 |

---

> > ### Comment · Reviewer_DyjW · 2024-11-27
> > **Response to Authors**
> >
> > Thanks to the authors for the response. I believe that the response and edits to the paper have addressed most of my concerns, and I have raised my score accordingly. One suggestion I have for the authors is to conclude the paper with a Conclusion section to more clearly outline the benefits of R2-Guard.

---

> > > ### Author Response · Authors · 2024-11-27
> > >
> > > Thank you for your valuable feedback! We have moved the Conclusion section from the Appendix to the end of the main text to emphasize it.

---

### Author Response · Authors · 2024-11-22
**Revision Summary**

We thank all the reviewers for their valuable comments and feedback! We are glad that the reviewers find our work solid and novel with sound empirical results. Based on the reviews, we have made the following updates to further improve our work.

1. We added more details on the evaluation setup and TwinSafety dataset in Sections 4 and 5, following the suggestion of Reviewer DyjW.

2. We added more analysis and empirical evidence of the advantage of $R^2$-Guard over ensemble learning, following the suggestion of Reviewer DyjW.

3. We improved the writing, such as providing an abstract overview figure and including illustrations of a running example in Section 3.1 for better understanding, following the suggestion of Reviewer t55i and Reviewer DyjW.

4. We included more discussions on the limitation of $R^2$-Guard in Section 5.3.3 and Appendix C, following the comment of Reviewer t55i.

5. We included more discussions on the application of $R^2$-Guard to open-world scenario in Section 5.3.3 and Appendix A.6, following the comment of Reviewer t9hR.

The updates in the revised manuscript are shown with highlighted color.

---

### Meta-Review · Area_Chair_K9yp · 2024-12-19

**Metareview:**

This paper introduces a novel approach to language model safety that combines probabilistic graphical models with traditional category-specific guardrails. The system implements additional reasoning through a PGM grounded in first-order logical rules, enabling it to capture relationships between different safety categories. The approach demonstrates superior performance compared to existing guardrails across multiple safety benchmarks, including significant improvements in resisting various types of attacks.

The authors also introduce a new benchmark designed to test guardrail resilience against more granular attacks and hybrid categories of harm. Despite its additional PGM component, the system maintains efficiency, showing similar latency to existing guardrails.

The work presents several technical strengths, including an innovative combination of classical PGM techniques with modern guardrails, offering an interpretable approach to content moderation through explicit logical rules. The system demonstrates strong empirical results across multiple benchmarks and shows significant improvements in resisting various attacks. It also shows strong potential for real-world applications due to its efficient performance, adaptability to new safety categories without full retraining, and interpretable decision-making process through explicit rules.

However, the paper has some limitations. It lacks detailed discussion of the process for creating and selecting logical rules, exploration of alternative rulesets, and justification for why such a simple ruleset yields significant improvements. There are also clarity issues regarding the pseudo-learning approach for PGM training and the relationship between ensemble-based training and final performance. The benchmark documentation could be improved with clearer methodology description and validation of dataset quality.

Despite these limitations, the paper merits acceptance based on its strong technical merit, practical impact, and value to the research community. The system presents a novel and effective approach to safety, combining classical and modern techniques innovatively. It shows immediate practical value through its efficiency, adaptability, and strong performance. While the paper would benefit from revisions addressing documentation and analysis gaps, its core contributions are significant enough to warrant acceptance.

**Additional Comments On Reviewer Discussion:**

The paper underwent significant review discussions focusing on various concerns raised by multiple reviewers. The reviewers highlighted the need for improved documentation and clarity in presenting technical concepts, evaluation setup details, and documentation of the dataset. Technical analysis concerns were raised regarding the training approach, the relationship between ensemble learning and system performance, and the need for more detailed category-specific performance analysis.

The reviewers also expressed concerns about handling edge cases, adaptability to new safety categories, and the need for a comprehensive discussion of system limitations. In response, the authors made substantial revisions to address these issues. They enhanced the documentation by adding detailed information about the evaluation setup and dataset, improved technical presentation with overview figures, and included illustrative running examples.

The authors provided additional analysis comparing system performance to ensemble learning approaches, expanded the discussion of system limitations, and added analysis of open-world scenarios. They also enhanced the explanation of their methodology and included more examples throughout the manuscript for better understanding.

These revisions significantly strengthened the paper by improving its accessibility through overview figures and running examples, enhancing empirical support through additional comparative analysis, and better defining the scope through expanded discussions of limitations and open-world scenarios. The changes effectively addressed the main concerns while preserving the paper's core contributions.

The authors' responsive and comprehensive revisions supported the decision to accept the paper, as they significantly improved its clarity and completeness without revealing fundamental flaws. Future research directions could include exploring alternative rulesets, conducting more extensive human evaluation studies, and deeper analysis of edge cases and failure modes, though these suggestions are not requirements for the current submission.

---

### Decision · Program_Chairs · 2025-01-22

Accept (Spotlight)